# Enhancement of Photodetector Characteristics by Zn-Porphyrin-Passivated MAPbBr_3_ Single Crystals

**DOI:** 10.3390/nano14131068

**Published:** 2024-06-21

**Authors:** Abdul Kareem Kalathil Soopy, Shengzhong (Frank) Liu, Adel Najar

**Affiliations:** 1Department of Physics, College of Science, United Arab Emirates University, Al Ain 15551, United Arab Emirates; 201990031@uaeu.ac.ae; 2Dalian National Laboratory for Clean Energy, Dalian Institute of Chemical Physics, Chinese Academy of Sciences, Center of Materials Science and Optoelectronics Engineering, University of Chinese Academy of Sciences, Dalian 116023, China; 3Key Laboratory of Applied Surface and Colloid Chemistry, Ministry of Education, Shaanxi Engineering Lab for Advanced Energy Technology, School of Materials Science and Engineering, Shaanxi Normal University, Xi’an 710119, China

**Keywords:** single crystals, passivation, photodetector

## Abstract

Perovskite single crystals have garnered significant interest in photodetector applications due to their exceptional optoelectronic properties. The outstanding crystalline quality of these materials further enhances their potential for efficient charge transport, making them promising candidates for next-generation photodetector devices. This article reports the synthesis of methyl ammonium lead bromide (MAPbBr_3_) perovskite single crystal (SC) via the inverse-temperature crystallization method. To further improve the performance of the photodetector, Zn-porphyrin (Zn-PP) was used as a passivating agent during the growth of SC. The optical characterization confirmed the enhancement of optical properties with Zn-PP passivation. On single-crystal surfaces, integrated photodetectors are fabricated, and their photodetection performances are evaluated. The results show that the single-crystalline photodetector passivated with 0.05% Zn-PP enhanced photodetection properties and rapid response speed. The photoelectric performance of the device, including its responsivity (R), external quantum efficiency (EQE), detective nature (D), and noise-equivalent power (NEP), showed an enhancement of the un-passivated devices. This development introduces a new potential to employ high-quality perovskite single-crystal-based devices for more advanced optoelectronics.

## 1. Introduction

The exceptional intrinsic properties of organic–inorganic hybrid perovskites CH_3_NH_3_PbX_3_ (X = Cl, Br, I) have propelled their remarkable success in the fields of photovoltaics and optoelectronics [1,2,3,4,5]. These properties encompass a high absorption coefficient, elevated carrier mobility, and prolonged carrier lifetime. Presently, most high-performance perovskite devices, including solar cells, transistors, and light-emitting diodes, are fabricated using multicrystalline thin films. However, the performance of these films may be constrained by the presence of numerous grain boundaries, voids, and surface defects [6,7]. Transitioning to single crystal-based devices is now recognized as a pivotal step in enhancing their performance, as single crystals boast exceptional crystallinity devoid of grain boundaries or morphological effects, which are advantageous for devices like photodetectors and X-ray imaging [8,9,10]. Therefore, acquiring single crystals of CH_3_NH_3_PbX_3_ with substantial dimensions and superior crystalline quality is imperative.

Among the several members of the CH_3_NH_3_PbX_3_ family, CH_3_NH_3_PbBr_3_ (MAPbBr_3_) stands out for its exceptional stability relative to MAPbI_3_ and its broader range of visible-light absorption compared to MAPbCl_3_. Moreover, MAPbBr_3_ features a cubic crystal symmetry, which promotes the growth of high-quality single crystals [11]. Numerous methods have been employed from the solution to grow MAPbBr_3_ single crystals since Dang et al. [12] started studying centimeter-sized MAPbI_3_ bulk crystals. These methods include solution cooling, top-seeded growth, and anti-solvent vapor-assisted crystallization (AVC), etc. [13,14,15,16]. Nevertheless, there are certain disadvantages to these reported perovskite single crystallization techniques, such as slow growth rates for large-scale applications or the fact that many crystals created with these solution procedures still require improvement in terms of crystallization quality for demanding optoelectronic applications [17]. The development of large-scale perovskite single crystals with outstanding quality and fast growth has the potential to broaden their applications. Using the inverse solubility of perovskite halides in particular solvents, the inverse temperature crystallization (ITC) method has become a widely used technique for producing large-scale, high-quality perovskite single crystals quickly [18,19,20]. This technique takes advantage of the solubility of perovskite molecules in the solvent at varying temperatures. The majority of the perovskite molecules in the ITC approach form complexes with the solvent, which lowers the number of unbound molecules, in contrast to conventional dissolving processes where perovskite molecules easily dissolve. These complexes maintain their stability at lower temperatures, preventing the solution from being saturated with free molecules. However, as the temperature increases, the binding energy of the complexes decreases. Eventually, the solution becomes supersaturated, causing the unbound perovskite molecules to rise and initiate crystallization [21].

Perovskite materials can contain defects and trap states that hinder the movement of charge carriers and induce charge carrier recombination, which affects their photovoltaic performance. By filling in surface dangling bonds and lowering the density of defects and trap states, researchers have attempted to address this problem using chemical passivation techniques [2,3,22]. Recently, porphyrin derivatives have surfaced as promising additives in perovskite optoelectronics [23,24] thanks to their notable light absorption characteristics, advantageous charge transport properties, and outstanding thermal stability. Bulk passivation, a technique entailing the comprehensive passivation of the crystal volume during its growth phase, presents distinct advantages over surface passivation strategies. This method facilitates superior control over growth defects, such as vacancies and dislocations, by mitigating their formation within the crystal lattice. Moreover, bulk passivation contributes to a reduction in trap density throughout the crystal structure, thereby enhancing charge carrier transport properties and ultimately improving the performance of optoelectronic devices [25].

In this study, we explore the effectiveness of zinc porphyrin (Zn-PP) as an example of passivating molecules during the growth of the MAPbBr_3_ single crystal. Large-scale and high-quality single crystals of MAPbBr_3_ were successfully produced using the inverse temperature crystallization (ITC) method. The percentage weight of Zn-PP was varied to optimize the growth, crystallinity, and phase of single crystals. Furthermore, we investigated the photodetectors based on MAPbBr_3_ SC, both with and without bulk passivation. We observed that 0.05 wt% of Zn-PP-based MAPbBr_3_ SC showed better photoelectric properties than pristine MAPbBr_3_ SC.

## 2. Materials and Methods

All materials used in this study were purchased from commercial sources and used as received without further purification. Lead bromide (PbBr_2_) and methyl ammonium bromide (MABr), N-N dimethylformamide (DMF), and chlorobenzene (CB) were purchased from Sigma-Aldrich. Zinc porphyrin (zinc(II) 5,10,15,20-(tetraphenyl)porphyrin) was purchased from PorphyChem (Longvic, France), and its chemical structure is exhibited in Appendix A.

MAPbBr_3_ single crystals were grown by the inverse temperature crystallization method [26]. Figure 1 shows the schematic of the step-by-step preparation of MAPbBr_3_ single crystals. In brief, 1M of MAPbBr_3_ growth solution was prepared by dissolving MABr and PbBr_2_ in a N-N dimethylformamide (DMF) solution before stirring continuously for 2 h until a clear, transparent solution was formed at room temperature with a molar ratio of 1:1. After dissolving completely, the solution was filtered using a 0.2 μm PTFE filter to remove any impurity particles derived from the raw materials and environmental atmosphere. Then, the transparent solution was divided among vials into 2 mL quantities and placed in an oil bath, and the oil temperature was maintained at 80 °C. For the preparation of Zn-PP-passivated MAPbBr_3_ SCs, different weight percentages of Zn-PP (0.05–0.15 wt%) with respect to PbBr_2_ were added to the growth solution. The crystals used for measurements were grown by transferring the seed crystals to a freshly prepared solution and repeating the crystallization procedure every 3 h to obtain SCs around 5 mm in size. All procedures were carried out under ambient conditions with a relative humidity of 55–60%.

Planar photodetectors were fabricated by depositing interdigital Au electrodes (100 nm in thickness) via the vacuum evaporation method on the top surface of the single crystals, with each electrode comprising a group of fine Au wires with a gap of 50 µm. The effective illuminated area of each photodetector was about 0.84 × 10^−3^ cm^2^.

Powder XRD patterns were used to analyze the phase and crystallinity of SC with a Rigaku X-ray diffractometer with a Cu Kα X-ray (λ = 1.54 Å) tube operated at 40 kV and 30 mA. The energy band gaps of MAPbBr_3_ single crystals were evaluated with the UV–Vis–NIR transmittance spectrum using a UV-Vis-NIR spectrophotometer (PerkinElmer Lambda 950, Waltham, MA, USA). Photoluminescence spectroscopy (PL) (NOST Co. Ltd., Seoul, Republic of Korea) was used to analyze the optical properties. The chemical bonding structure was characterized by Fourier transform infrared spectroscopy (FTIR). The photocurrents of these photodetectors under various light intensities were measured by using a Keithley 4200 semiconductor characterization system and a manual probe station; a 405 nm semiconductor laser was used as the light source. To measure the response time, a 532 nm semiconductor laser driven by a signal generator (Tektronix, AFG3252C, Beaverton, OR, USA) as the light source was used to generate a pulsed laser beam, and the temporal response photocurrent of the device was measured using a low noise current preamplifier (Stanford Research System, SR570, Sunnyvale, CA, USA) with a Mixed Domain Oscilloscope (Tektronix, MDO3104, Beaverton, OR, USA).

## 3. Results and Discussion

Crystallization in hybrid halide perovskites is initiated by a temperature-dependent reverse solubility phenomenon in certain solvents. Unlike the usual dissolution process, most perovskite molecules interact with the solvent to form complexes, impeding dissolution. Additionally, unbound molecules do not achieve low-temperature saturation levels. As temperature increases, the binding energy of these complexes decreases, resulting in a greater number of unbound perovskite molecules. Consequently, nucleation occurs when the solution reaches supersaturation, followed by crystal growth [18,21]. We observed the quick formation of tiny MAPbBr_3_ perovskite precipitates at elevated temperatures (80 °C) and in the concentrated precursor solution (1M) containing equal amounts of lead bromide (PbBr_2_) and methylammonium bromide (MABr). However, through careful control of temperature and precursor concentration in DMF, we managed to generate only a limited number of crystals. Figure 2a shows a photograph of the MAPbBr_3_ SCs fabricated by the inverse temperature crystallization method. MAPbBr_3_ single crystals of about 0.5 × 0.5 cm in size were achieved after 12 h of crystallization reaction. For the passivation of MAPbBr_3_, Zn-PPs were used as passivating molecules during the growth process. MAPbBr_3_ SCs were passivated with three different weight percentages of Zn-PP (0.01, 0.05, 0.15 wt%). Notably, we found that for both MAPbBr_3_ single crystals and Zn-PP-passivated MAPbBr_3_ single crystals, the crystallization process was reversible. Upon cooling the crystals back to room temperature in the growth solution, they dissolved, illustrating the reversible characteristic of the crystallization process. We observed that MAPbBr_3_ single crystals did not undergo any shape changes after using bulk Zn-PP passivation during the growth of SC, and their color remained approximately the same.

To identify the phase and crystallinity, MAPbBr_3_ SC was characterized by an XRD spectrometer. Figure 2b presents the XRD results of MAPbBr_3_ SCs with the passivation of different Zn-PP weight percentages. The results indicate that there are four main diffraction peaks for the reference sample, as shown in Figure 2b, which appeared at around 15.78, 30.94, 46.64, and 63.31 and were related to [100], [200], [300], and [400] crystal planes, respectively, which confirmed that all the MAPbBr_3_ single crystals possessed cubic crystal structure, as reported previously by different groups [27,28,29,30]. Furthermore, the absence of discernible impurity peaks in the results obtained from the pristine MAPbBr_3_ single crystals suggests the high purity of the crystals grown via the ITC method. This observation underscores the effectiveness of the growth technique in yielding pristine MAPbBr_3_ single crystals devoid of significant impurities. The observed increase in [100] plane intensity following passivation indicates a notable structural enhancement in the MAPbBr_3_ single crystal. Specifically, when the single crystal was passivated with 0.05% Zn-PP, the intensities of the [100] and [200] planes became comparable, suggesting a balanced improvement in the crystal’s orientation along both planes. This observation suggests that the presence of 0.05% Zn-PP might play a crucial role in promoting the growth of the perovskite single-crystal along the [200] direction. This facilitation of growth could lead to an improvement in the structural quality of the perovskite single crystal, potentially resulting from improved passivation, which is often associated with improved material quality, reduced defect density, and enhanced electronic properties. However, when the weight percentage of Zn-PP exceeds 0.05%, the emergence of impurity phases within the MAPbBr_3_ single crystal is observed. The presence of impurity phases suggests that higher concentrations of Zn-PP may induce non-ideal crystallization conditions or chemical interactions that disrupt the integrity of the perovskite structure. Therefore, while 0.05% Zn-PP passivation appears to effectively enhance the crystal quality of MAPbBr_3_ single crystals, excessive concentrations can lead to unintended structural modifications and impurity formation. Therefore, we suggested that Zn-PP passivation with a high weight percentage could have an impact on the MAPbBr_3_ crystal’s crystallization quality.

The UV–Vis–NIR transmittance spectra of MAPbBr_3_ single crystals, both with and without Zn-PP passivation, are displayed in Figure 3a,b along with the fitting results, respectively. The measured wavelength ranged from 300 to 1000 nm. As shown in Figure 3a, the transmittance of Zn-PP-passivated MAPbBr_3_ single crystals decreased gradually at wavelengths greater than 560 nm. However, the transmittance curves can be broadly classified into three regions: (1) a strong absorption region, where the incident light wavelength is less than 560 nm; (2) an exponential absorption region, where the incident light wavelength ranges from 560 to 590 nm; and a (3) weak absorption region, where the incident light wavelength exceeds 590 nm. However, the 0.15% Zn-PP sample does not follow this specific trend, which may be due to the non-uniform distribution or incorporation of Zn-PP at this specific concentration. Additionally, the spectral features observed at wavelengths greater than 560 nm, particularly the band around 900 nm, can be attributed to sub-band gap states or defect states introduced by Zn-PP passivation. These features suggest complex interactions between Zn-PP and the MAPbBr3 crystal lattice, which do not exhibit a straightforward trend with the increasing Zn-PP concentration.

The Tauc plot analysis, as represented in Figure 3b, revealed that the un-passivated sample has a band gap of 2.193 eV. Conversely, the Zn-PP-passivated samples exhibit a reduced band gap, measured at 2.181 eV. Notably, this value aligns more closely with the reported band gap of 2.17 eV for single crystals [31], although it remains below the band gap range observed in films (2.30 ± 10 eV) [32,33]. In optoelectronic applications covering a broader spectrum of light, a narrower band gap (E_g_) would be beneficial. The graphical representation in the inset of Figure 3b shows how the passivation process affects the band gap energy of the single crystals. This representation offers valuable insight into the structural and optical changes induced by passivating agents. Initially, as the weight percentage of zinc porphyrin increases, more Zn-PP molecules are available to passivate the defects, reducing non-radiative recombination and enhancing absorbance while decreasing the effective band gap energy. However, when the weight percentage of Zn-PP further increases to 0.15%, the formation of aggregates or clusters on the material’s surface may occur, introducing new energy levels and altering the electronic structure, leading to an increase in band gap energy and a decline in absorbance due to increased light scattering. This observation is consistent with the presence of impurity phases observed in the XRD spectrum depicted in Figure 2b when the weight percentage of Zn-PP increased to more than 0.05%.

The photoluminescence performance of MAPbBr_3_ single crystals was examined using steady-state photoluminescence (PL) spectra with an excitation light of 532 nm. Figure 4a shows the room temperature PL spectra of passivated MAPbBr_3_ SC with different weight percentages of Zn-PP collected from the top of single crystals. The PL peak of the MAPbBr_3_ single crystal without any Zn-PP passivation is located at 536 nm, and it agrees with the previous literature [34,35]. However, we noticed a 6 nm redshift in the Zn-PP-passivated samples with an increase in the PL intensity as the passivation weight percentage increased. The red-shifted and narrower PL peak indicates the single crystal can be attributed to a highly ordered structure and low-defect density [36,37,38]. Indeed, this observation is consistent with the decrease in E_g_ when the passivation weight percentage increased, as observed in Figure 3b. Therefore, it is evident that the surface as well as bulk defect states decreased with the addition of Zn-PP during the growth of MAPbBr_3_ SC.

The optimal performance observed at 0.05% Zn-PP can be attributed to effective defect passivation, as indicated by the significant increase in photoluminescence (PL) intensity, which suggests reduced non-radiative recombination. Beyond this concentration, performance metrics deteriorated due to the aggregation of Zn-PP molecules, which introduced new defects and disrupted the crystal lattice, as evidenced by the decrease in PL intensity. These aggregations likely impeded charge transport, resulting in increased resistance and reduced mobility, thereby outweighing the benefits of additional passivation.

The FTIR spectrum of both the passivated and un-passivated MAPbBr_3_ SC samples exhibited infrared (IR) peaks at 3185 cm^−1^ and 3148 cm^−1^, as shown in Figure 4b. These peaks can be ascribed to the split N–H symmetric stretching mode of the methylammonium (MA^+^) cation [39,40]. Additionally, all the FTIR spectra in Figure 4b show almost identical bands with C–H vibrational modes at 2930 cm^−1^ and C–N stretching modes at 1248 cm^−1^, as reported in other studies in the literature [41,42,43]. The peaks observed at approximately 912 cm^−1^ and 966 cm^−1^ can be attributed to the CH_3_–NH_3_^+^ rocking mode and C–N stretching mode, respectively [44]. Additionally, peaks appearing around 1473 cm^−1^ and 1571 cm^−1^ are identified as corresponding to the NH_3_^+^ symmetric bending mode and NH_3_^+^ asymmetric bending mode, respectively [45]. These IR characteristics indicate that the inorganic perovskite structures contain organic MA^+^ cations. The peaks observed within the wavenumber range of 2844–2973 cm^−1^ can be assigned to C–H stretching vibrational modes, while those within the range of 3000–3300 cm^−1^ correspond to N–H stretching vibrational modes. Furthermore, the peak found in the wavenumber range of 3300–3600 cm^−1^ is attributed to O–H vibrations [46]. The additional peak that appeared at 811 cm^−1^ and 1004 cm^−1^ can be assigned to Zn-PP-passivated MAPbBr_3_, which confirms the presence of the passivating molecules effect in single-crystal formation [47,48,49].

A planar-type photodetector (PD) based on MAPbBr_3_ single crystals with and without Zn-PP was fabricated as described in the experimental part. Figure 5a illustrates the magnified picture of four photodetectors fabricated on the surface of MAPbBr_3_ single crystal on the perovskite facet. Every detector comprises a pair of interdigitated wire electrodes, where each electrode is composed of fine Au metal wires with a thickness of approximately 100 nm. The gap between each electrode measures around 50 μm, and the effective illuminated area for each photodetector is approximately 0.84 × 10^−3^ cm^2^. The schematic diagram and device architecture of the MAPbB_3_ single crystal photodetector device for the photo-response measurement are displayed in Figure 5b. A 405 nm semiconductor laser was used as the light source for the photocurrent measurements of these PDs, and a 532 nm semiconductor laser was used as the light source to generate a pulsed laser beam for the response time measurements.

To assess the photoelectric performance of the device, we analyzed the important characteristics of photodetectors, such as responsivity (R), external quantum efficiency (EQE), and detective level (D). Responsivity (R) is defined as the generated photocurrent per unit power when incident light illuminates the effective area of the detector. The responsivity R can be calculated using the following equation [50,51].
(1)$$R=\frac{I_{light}-I_{dark}}{PA}$$
where I_light_ and I_dark_ are the photocurrents under light, and in the dark, P is the illumination power density, and A is the effective area of the illumination.

The number of electrons detected per incident photon is known as the external quantum efficiency (EQE), and it can be computed using the following relation [52].
(2)$$EQE=\frac{Rhc}{q\lambda }$$
where R is the responsivity, h is Planck’s constant, c is the light velocity, λ is the wavelength of incident light, and q is the electron charge.

The detective quality of the photodetector is calculated using the following equation [53]:(3)$$D=R/\sqrt{2qI_{0}}$$
where R_λ_ is the responsivity, q is the charge of electrons, and I_0_ is the dark current, representing the current that flows through the photodetector in the absence of incident light.

Noise-equivalent power (NEP), which is a measure of the sensitivity of a photodetector, is calculated using the following equation:(4)$$NEP=\frac{\sqrt{A∆f}}{D^{*}}$$
where A is the effective area, Δf = 1 Hz, and D* is the specific detective level, which is defined as the detective level D for 1 Hz bandwidth and a 1 cm^2^ area.

The results of the device parameters are summarized in Table 1.

The dark current plays a crucial role in shaping the performance characteristics of photodetectors. Its presence can lead to a notable impact on the device’s operational efficiency. The dark current essentially represents the electrical current that flows through a photodetector in the absence of any incident light. This background current reduces the device’s light conversion efficiency by introducing a baseline signal that can interfere with the detection of light-induced currents. Moreover, the presence of a dark current amplifies the overall noise level in the system, which can degrade the signal-to-noise ratio and compromise the detector’s sensitivity and accuracy. Therefore, minimizing dark current is imperative for optimizing the performance and reliability of photodetectors in various applications [54,55]. The photocurrent of the devices fabricated with different Zn-PP levels of passivation has been measured under dark conditions, and the results are shown in Appendix A. As observed, the Zn-PP-passivated photodetectors (PDs) exhibited a more limited dark current compared to pristine MAPbBr_3_ single crystals (SCs), a characteristic advantageous for photodetector applications. Thus, it is demonstrated that Zn-PP passivation can effectively inhibit the trapping on the perovskite surface to enhance the photodetector performance. The dark current is the minimum for the Zn-PP 0.05%-passivated MAPbBr_3_ SC. The current–voltage curves (I–V curve) of MAPbBr_3_ single crystals were measured under illumination using a 405 nm LED light with different light intensities and are depicted in Appendix A. It is noted that the photocurrent increases with applied voltage. The photodetector shows a dark current as low as ~10^−6^ A, while the photocurrent increases to ~10^−5^ A under illumination (10.8 mW cm^−2^ at 405 nm). However, the non-linear and asymmetrical behaviors observed in the I–V curve are likely attributed to the voltage-induced ion drift [19,56].

The responsivity (R), EQE, detective rate (D), and noise equivalent power (NEP) of the single crystals fabricated with different weight percentages of Zn-PP are calculated according to Equations (1), (2), (3) and (4), respectively. Responsivity and EQE rely on bias voltages and illumination power. Table 1 summarizes the photodetection performance of MAPbBr_3_ single crystals with different weight percentages of Zn-PP with a bias voltage of 5 V and illumination power of 47.5 mW cm^−2^. The sample passivated with 0.05% Zn-PP exhibited the best performance among these with a responsivity of 5.16 (A/W) and EQE of about 1581%, whereas the un-passivated sample exhibited a responsivity of 1.05 (A/W) and EQE of 320%. Our device’s performance is comparable to previously reported MAPbX_3_-based photodetectors. Liu et al. achieved an impressive responsivity and detective rate of 62.9 A/W and 6.5 × 10^12^ Jones, respectively, using a micro concave photodetector with a 520 nm light with 3 V bias, although their device structure was considerably more complex [31]. In another study, Au/MAPbBr_3_/Au photodetectors demonstrated a responsivity of 16 A/W, EQE of 3900, and a detective ability of 6 × 10^13^ [57]. Additionally, self-powered photodetectors were developed in CH_3_NH_3_PbI_3_-CH_3_NH_3_PbBr_3_ thin film heterojunctions. However, it is worth noting that their device architectures differ from our photodetector [58]. A detailed comparison between the published results and ours is provided in the Appendix A. The low trap density and high carrier mobility achieved upon passivation using Zn-PP could be responsible for the higher responsivity and EQE, making it more efficient in converting light into electrical signals [59,60].

In most photodetection applications, the consistent and dependable performance of a photodetector across a wide range of light intensities is essential. Figure 6a,b depict the photocurrent density and responsivity obtained for the pristine and Zn-PP-passivated MAPbBr_3_ SC-based photodetectors using a 405 nm LED with different light intensities under a 10 V bias voltage, respectively. It is shown that the detective and responsive values are higher for the passivated samples. Also, the responsivity decreases with increases in the light intensity. The highest overall performance parameters were obtained for the Zn-PP 0.05%-passivated sample as expected. This enhancement in photodetector properties confirmed the reduced trap state densities for the passivated samples compared to the pristine device, which agreed with the enhanced PL observed. The photo response speed is a significant characteristic of photodetectors, and it is directly associated with the extraction of photogenerated charge carriers. In Figure 6c, a promising demonstration of device performance is presented. The device response time, denoted as t_r_ and t_f_, can be determined by estimating the duration taken for the photocurrent to increase from 10% to 90% and decrease from 90% to 10%, respectively [61]. Thanks to their high structural quality and low defect density, the photo response of Zn-PP-passivated single-crystal photodetectors was faster than the control device. The photocurrent for the un-passivated sample showed a t_r_ of 7.5 μs and t_f_ of 84.8 μs as a function of time. However, with a 0.05% Zn-PP passivation, the MAPbBr_3_ SC performed with a reduced t_r_ of 2.6 μs and a t_f_ of 57.9 μs. These photo-response speeds are approximately 10^3^ times higher than those reported for photodetectors based on CH_3_NH_3_PbBr_3_ nanowires (t_r_ = 120 ms and t_f_ = 86 ms) [62] and CH_3_NH_3_PbI_3_-CH_3_NH_3_PbBr_3_ heterojunctions (t_r_ = 120 ms and t_f_ = 94 ms) [58]. Utilizing PEDOT:PSS and PCBM as charge-transporting layers, Dou et al. reported that their perovskite photodetectors could operate in the MHz regime [63]. Compared with other perovskite-film-based photodetectors that do not utilize electron/hole transporting layers [64,65], the significantly faster photo-response of our devices could be attributed to the high carrier mobility and low defect traps inherent in the perovskite single crystals. Based on the above enhanced photo-detecting properties, it is expected that the Zn-PP-passivated MAPbBr_3_ single crystalline perovskite is a promising candidate for high-performance optoelectronics.

## 4. Conclusions

In summary, we successfully synthesized single-crystalline CH_3_NH_3_PbBr_3_ perovskites using a facile solution method, with zinc porphyrin (Zn-PP) serving as the bulk passivation agent. Remarkably, the 0.05% Zn-PP-passivated samples exhibited superior optical and electrical properties, as indicated by their more intense and narrower photoluminescence (PL) peaks, suggesting their effective control over defect states. Furthermore, the photodetector performance metrics, including responsivity and external quantum efficiency (EQE), were notably enhanced in the 0.05% Zn-PP-passivated MAPbBr_3_ single crystals. These findings underscore the significant potential of Zn-PP passivation in improving device performance. It is expected that this passivation method will bring large perovskite single crystals to the market, and their availability may revolutionize the broad application of optoelectronics.

## Figures and Tables

**Figure 1 nanomaterials-14-01068-f001:**
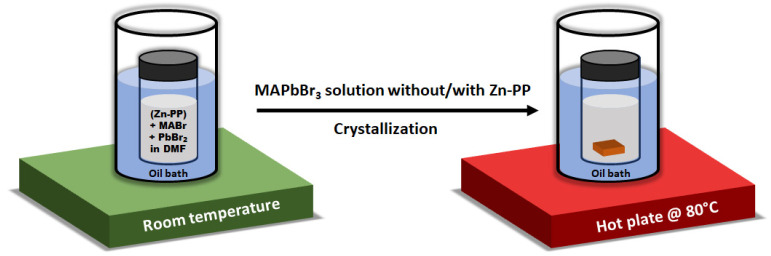
Synthesis schematic of MAPbBr_3_ single crystal without or with zinc porphyrin using inverse temperature crystallization method.

**Figure 2 nanomaterials-14-01068-f002:**
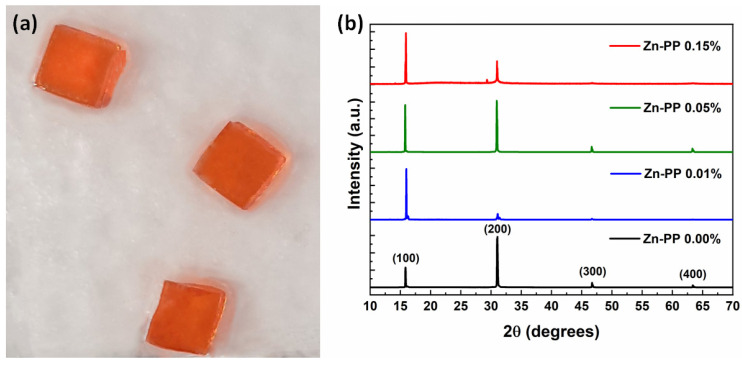
(**a**) Photograph of as-synthesized MAPbBr_3_ single crystals. (**b**) XRD patterns of MAPbBr_3_ single crystals prepared by inverse temperature crystallization method with different weight percentages of Zn-PP.

**Figure 3 nanomaterials-14-01068-f003:**
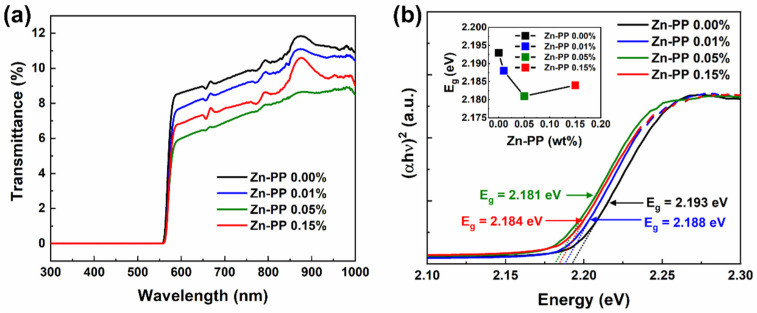
(**a**) UV-Vis-NIR transmittance spectra of MAPbBr_3_ single crystals without/with Zn-PP passivation. (**b**) and their Tauc plot. The inset shows the band gap energy (E_g_) vs. Zn-PP passivation.

**Figure 4 nanomaterials-14-01068-f004:**
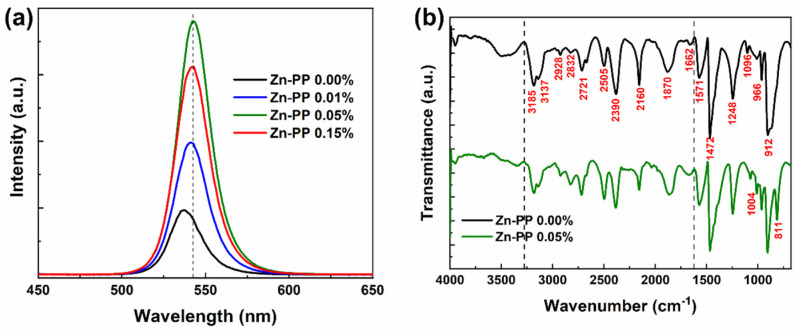
(**a**) Photoluminescence spectrum of the MAPbBr_3_ single crystals without/with zinc porphyrin passivation excited at 532 nm. (**b**) FTIR spectra of MAPbBr_3_ single crystal without and with zinc porphyrin.

**Figure 5 nanomaterials-14-01068-f005:**
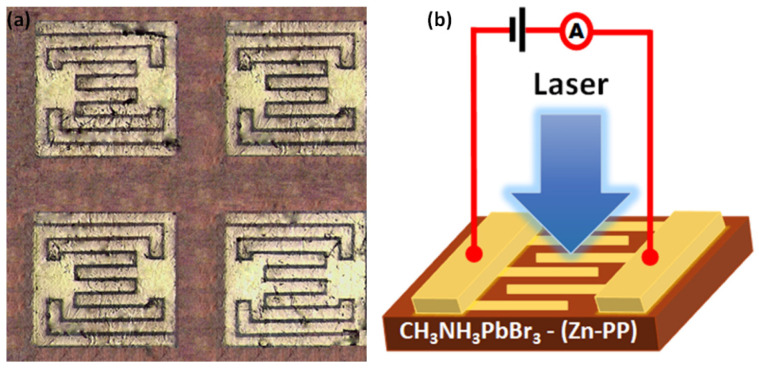
(**a**) A photograph of a planar-type photodetector fabricated on MAPbBr_3_ single crystal. (**b**) Device architecture of the MAPbBr_3_ single crystal planar-type photodetector.

**Figure 6 nanomaterials-14-01068-f006:**
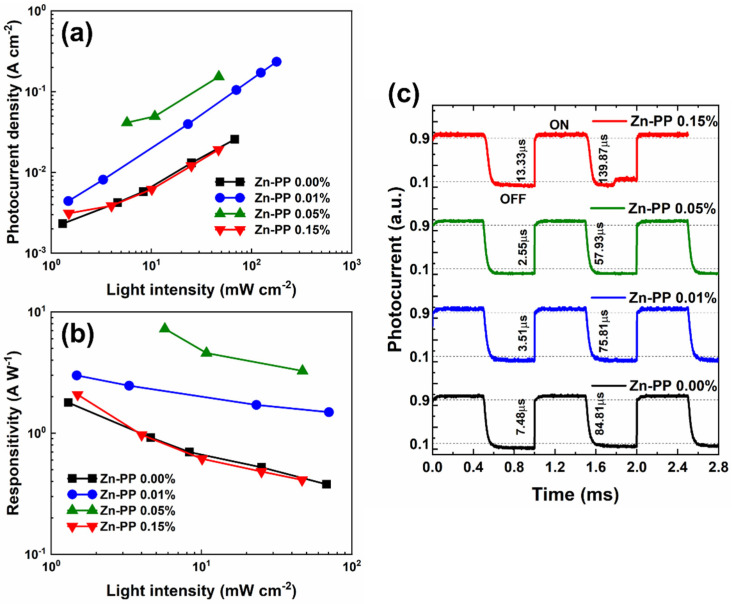
(**a**) Photocurrent density versus optical power density plot of pristine and Zn-PP-passivated MAPbBr_3_ SCs under 10 V bias. (**b**) Responsivity versus optical power density plot of pristine and Zn-PP-passivated MAPbBr_3_ SCs under 10 V bias. (**c**) Transient photo response of fabricated photodetectors.

**Table 1 nanomaterials-14-01068-t001:** The photodetector performance of fabricated MAPbBr3 single crystal devices with different levels of passivation.

Device	Responsivity (AW^−1^)	EQE (%)	Detectivity (Jones)	NEP (WHz^1/2^)
Zn-PP 0.00%	1.05	320	8.08 × 10^11^	1.24 × 10^−12^
Zn-PP-0.01%	2.22	679	4.24 × 10^12^	2.36 × 10^−13^
Zn-PP-0.05%	5.16	1581	4.76 × 10^12^	2.10 × 10^−13^
Zn-PP-0.15%	4.01	1227	1.33 × 10^12^	7.54 × 10^−13^

## Data Availability

Data are contained within the article and Appendix A.

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
