# Peer review of "Enhancement of Photodetector Characteristics by Zn-Porphyrin-Passivated MAPbBr3 Single Crystals"

_nanomaterials, 2024, doi:10.3390/nano14131068_

Round 1

Reviewer 1 Report

Comments and Suggestions for Authors

In the work “Enhancement of Photodetector Characteristics by Zn-Porphyrin Passivated MAPbBr3 Single Crystals”, Soopy and co-authors report hybrid organic-inorganic perovskite single crystals grown via inverse-temperature crystallization with different amounts of a Zn-porphyrin bulk passivating agent. The authors demonstrate that an optimised amount of passivating agent improves both the quality of the crystals and the photoelectric performance of photodetectors based on the said crystals. The authors’ approach to improve the performance of perovskite photodetectors via passivation is not novel but has the potential to significantly enhance the performance of devices based on perovskite single-crystal photodetectors.

Some revisions are needed, as detailed in the comments below:

1. Abstract, line 19; Intro, line 82: What Zn-porphyrin derivative are the authors referring to? The extended chemical name should be reported, at least in the Materials and methods section.

2. Results and discussion, line 176. Square brackets should be used to indicate crystallographic directions. And, what is there a reason for suggesting that the 0.05% concentration promotes the growth along the [200] direction, and not the [100]. Are they not equivalent?

3. Line 195. “the transmittance of Zn-PP passivated MAPbBr3 single crystals decreases gradually in the wavelength less than 560 nm”. In Figure 3a the gradual drop occurs at wavelengths higher than 560 nm, not lower. Also, it seems that the 0.15% Zn-PP sample does not follow the trend. Finally, can authors comment on the spectral features at wavelength > 560 nm in Figure 3a? What is their spectroscopic origin? For example, the band at about 900 nm does not seem to follow a specific trend at increasing Zn-PP concentration.

4. Page 7. Authors opted to include all the relevant equations describing the device characteristics. All quantities should be defined, including D* and I0, whose definition is missing in the text.

5. Lines 316-319. To which photodetector are the authors referring to? The undoped one? And why are they quoting only the 10.8 mW cm-2 light intensity? Authors should double-check the current and photocurrent values reported/clarify at what bias voltage they are referring to.

Author Response

Dear Reviewer,

Thank you very much for your insightful comments and we are very grateful for your thorough evaluation and valuable suggestions. We have thus taken ALL suggestions into account, and the manuscript was revised in detail to address every point raised by the reviewer and all the modifications were highlighted in red in the manuscript and in the Supporting Information. Please find our point-by-point response in the attached file.

Reviewer 2 Report

Comments and Suggestions for Authors

The study focuses on the synthesis and application of perovskite single crystals, particularly methyl ammonium lead bromide (MAPbBr3), for photodetector devices. It explores the use of Zn-porphyrin (Zn-PP) as a passivating agent to enhance the optical and photoelectric properties of these materials. The research demonstrates that Zn-PP passivation significantly improves the performance metrics such as responsivity, external quantum efficiency, and detectivity of investigated material. It was found that while 0.05% Zn-PP passivation appears to effectively enhance the crystal quality of MAPbBr3 single crystals, excessive concentrations can lead to unintended structural modifications and impurity formation. The influence of Zn-PP molecules on the bandgap of the obtained samples has been studied. The presence of the passivating molecules on the surface of single crystals was checked using FTIR spectroscopy, and some additional peaks were found. A planar-type photodetector was fabricated on the surface of the MAPbBr3 single crystal.

It seems like I have no questions about the manuscript, but I do have a question regarding the choice of journal. Why Nanomaterials? The only nanoscale element I noticed was the Au electrodes with a thickness of 100 nm.

Author Response

(The authors gave the same response as above.)

Reviewer 3 Report

Comments and Suggestions for Authors

This manuscript is devoted to studying the possibilities of improving the performance of a photodetector based on (MAPbBr 3 ) perovskite single crystal through the use of Zn-porphyrin (Zn-PP) as a passivating agent during the growth of crystals. The authors found the optimal level of Zn-PP content, which provides significant increase in responsivity and External Quantum Efficiency of sample photodetector structures. The article may be of interest to specialists in the field of solar energy, photonics and solid state physics.

Notes on the manuscript.

It may seem to the reader that the higher the level of passivating agent content, the lower the density of defects and trap states in the perovskite crystal and, therefore, the higher the photodetector performance metrics should be tending to their best limiting value. However, these performance metrics are best when using 0.05% Zn-PP passivated samples and deteriorate with a further increase in the content of the passivating agent. The reasons for this effect need to be discussed.

In expression (3), the value D (detectivity of the photodetector) does not have the “*” sign. And in expression (4) it already has this sign. Do you mean complex conjugation? Or is it still a typo?

After expression (3) there is a signature “Jones” in parentheses. It is clear that the detectivity of the photodetector is measured in these units. But why indicate the dimension after the expression (and only for one formula)? It is sufficient that this dimension is indicated in Table I.

Author Response

(The authors gave the same response as above.)
